# Toward Efficient Inference Attacks:
# Shadow Model Sharing via Mixture-of-Experts

**Li Bai[1], Qingqing Ye[1], Xinwei Zhang[1], Sen Zhang[1], Zi Liang[1], Jianliang Xu[3], Haibo Hu[1,2]***
Department of Electrical and Electronic Engineering, The Hong Kong Polytechnic University[1]
PolyU Research Centre for Privacy and Security Technologies in Future Smart Systems[2]
Department of Computer Science, Hong Kong Baptist University[3]
`baili.bai@connect.polyu.hk, haibo.hu@polyu.edu.hk`

## Abstract

Machine learning models are often vulnerable to inference attacks that expose sensitive information from their training data. Shadow model technique is commonly employed in such attacks, such as membership inference. However, the need for a large number of shadow models leads to high computational costs, limiting their practical applicability. Such inefficiency mainly stems from the independent training and use of these shadow models. To address this issue, we present a novel shadow pool training framework *SHAPOOL*, which constructs multiple shared models and trains them jointly within a single process. In particular, we leverage the Mixture-of-Experts mechanism as the shadow pool to interconnect individual models, enabling them to share some sub-networks and thereby improving efficiency. To ensure the shared models closely resemble independent models and serve as effective substitutes, we introduce three novel modules: path-choice routing, pathway regularization, and pathway alignment. These modules guarantee random data allocation for pathway learning, promote diversity among shared models, and maintain consistency with target models. We evaluate SHAPOOL in the context of various membership inference attacks and show that it significantly reduces the computational cost of shadow model construction while maintaining comparable attack performance.

## 1 Introduction

With the increase of machine learning models trained on sensitive personal data like facial images and medical records, various inference attacks pose severe threats to them, such as inferring the membership status of individual records (i.e., membership inference attacks) [1, 2, 3], reconstructing representative samples (i.e., model inversion attacks) [4, 5], and uncovering statistical properties of the training dataset (i.e., property inference attacks) [6, 7].

A common technique adopted in almost all such attacks is the shadow model technique, which enables an adversary to train multiple models using the same architecture as the target model on similar datasets. Recent works show it is very effective in membership inference [1, 8, 9, 10] and property inference attacks [11, 12, 13, 14]. Typically, the number of shadow models determines the quality of inference attacks. For example, to achieve high attack performance, this number is set to 256 and 100 in [9, 11], respectively. However, the high computational cost of training shadow models has raised concerns [15, 16, 14]. Several works based on shadow models attempt to address this issue by optimizing this cost inside of their specific attack algorithms. For instance, QMIA [17] focuses solely on non-member data to develop a quantile regression model, thereby eliminating the

---

*Corresponding author

need for shadow models from other distributions. RMIA [16] employs a pre-trained reference model on population data to save the training cost for shadow models in non-member cases. SNAP [14] reduces the number of shadow models required by injecting poisoned samples that contain the target properties of interest for property inference. However, since these methods are tightly coupled with specific algorithms, they lack generalizability across different inference attacks.

In this paper, we address this problem from the perspective of shadow models themselves, that is, training each shadow model independently is not efficient [1, 13]. Inspired by this, we present *SHAPOOL*, a shadow pool training framework that constructs **shared shadow models** in a single training process. Specifically, we leverage the popular Mixture-of-Experts (MoE) mechanism [18, 19, 20] to build the pool with multiple identical sub-networks (i.e., experts) during training, and then generate shared models (i.e., activated pathways) as a substitute for conventional shadow models during inference. However, a vanilla MoE may produce over-specialized, unstable, and under-trained experts, which reduces the diversity of shared models, poorly resembles independent models, and ultimately diminishes attack effectiveness. To overcome these challenges, we design three key modules: pathway-choice routing, pathway regularization, and pathway alignment. The path-choice routing strategy assigns inputs to specific pathways to ensure a randomized yet fixed data allocation for pathway learning on top of stability and reliability. Pathway regularization aims to enhance the diversity of shared models by enforcing orthogonality in the representation space of experts and penalizing paired pathways with similar outputs. Lastly, pathway alignment is introduced to improve consistency with independent models and mitigate generalization mismatches.

We conduct extensive experiments to evaluate SHAPOOL across various existing membership inference attacks (MIAs) [9, 16], demonstrating that it outperforms the conventional shadow models in both attack performance and training efficiency. On one hand, in a resource-limited scenario where only a limited number of shadow models can be trained [21], shared models can enhance the performance of existing attacks under a similar computational budget. On the other hand, in resource-abundant scenarios like data auditing and risk assessment [22, 23, 24], it achieves comparable performance while significantly reducing the computational cost of conventional methods.

Overall, our contributions are as follows: (1) We present a novel shadow pool training framework. To the best of our knowledge, this is the first work to enhance the efficiency of various inference attacks from the perspective of the shadow model construction. (2) We propose utilizing the MoE mechanism as a shadow pool to interconnect individual models, enabling them to share sub-networks and enhance the overall training efficiency. (3) To ensure that shared models serve as effective substitutes, we introduce three modules that guarantee specialized data allocation for pathway learning, promote diversity among shared models, and ensure consistency with independent models. (4) Extensive experiments demonstrate the effectiveness and efficiency of our proposed method compared to conventional shadow models.

## 2 Preliminary

### 2.1 Shadow Model-based Inference Attacks

Shadow model technique is a widely-used approach for various inference attacks [1, 8, 9, 10, 11, 12, 13, 14]. Typically, an adversary constructs $n$ diverse shadow models $\{M_{S_1}, \ldots, M_{S_n}\}$ to simulate the behavior of the target model $M_T$. The $i$-th shadow model $M_{S_i}$ is often trained using the same architecture as $M_T$ on a similar auxiliary dataset $D_{A_i}$, and provides attack training data and corresponding ground-truth labels for the construction of the attack model $\mathcal{A}$. For example, in MIAs, shadow models generate outputs (e.g., logits or confidence scores) along with their corresponding membership labels (1 for member and 0 for non-member), which are then used to construct the attack training dataset for $\mathcal{A}$ (e.g., a binary classifier) to predict membership status. The pseudocode for shadow model-based inference attacks is provided in Appendix A.1, where more related works are discussed. To ensure effective inference attacks, a large number of shadow models are typically required to capture the diversity and randomness inherent in the training data distribution [17]. This leads to considerable computational overhead, especially for large-scale models, thereby limiting the practicality of shadow model-based attacks. In this paper, we introduce a novel training framework to advance the efficiency of shadow model construction.

## 2.2 Mixture-of-Experts Mechanism

A vanilla MoE model consists of $L$ expert layers, each containing $M$ experts with identical network structures and a router $\mathcal{R}$. Given an input $x$, the output of an expert layer is computed as the summation of the outputs from the top $K$ experts selected by $\mathcal{R}$:

$$h = \sum_{k=1}^{K} \mathcal{R}(x) \cdot E_k(x), \ \mathcal{R}(x) = \text{TopK}(\mathcal{G}(x), K), \tag{1}$$

where $E$ is a learnable expert, $\mathcal{G}$ is a routing strategy based on trainable networks or heuristic functions, and $\text{TopK}(\cdot, K)$ refers to the largest $K$ values. With $K = 1$ and non-expert layers omitted, we obtain an activated end-to-end network pathway, formally defined as a sequence of selected experts across layers, i.e., $\{E_i^1, E_j^2, ..., E_k^L\}$, where $E_i^1$ indicates that the $i$-th expert in the first layer is activated and $1 \leq i, j, k \leq M$. A more detailed introduction to the MoE architecture can be found in Appendix A.2.

## 3 Analysis of Shadow Models

Ideally, an inference attack should be both effective (i.e., high inference accuracy) and efficient (i.e., computationally cheap). We investigate which aspects of shadow models influence attack effectiveness via a case study on MIA, providing guidance for optimizing their construction.

We at first introduce a simple approach, model augmentation [25, 26, 27, 28, 29], to accelerate shadow model construction. For each trained shadow model, we generate an augmented version, doubling the total number of shadow models while reducing the overall construction cost, detailed in Appendix B. Specifically, we use neural masking [28, 30] that randomly prunes the weights of a base model's parameters in fully connected (*FC*) or convolutional (*Conv*) layers with a predefined probability $p$. However, unfortunately, those attacks with augmented models fall short of expectations in Figure 1(a). Next, we explore why augmented models are not effective from three perspectives: consistency, diversity, and randomness.

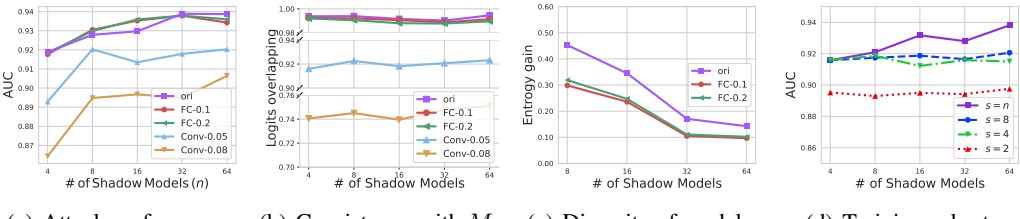

(a) Attack performance.  (b) Consistency with $M_T$.  (c) Diversity of models.  (d) Training subsets.

Figure 1: (a): Attack performance and (b-d): Factors of shadow models. We evaluate augmented shadow models on LiRA [9]. *ori* denotes attacks using conventional shadow models. *FC-p* (or *Conv-p*) denotes attacks with augmented models created via neural masking applied to fully connected layers (or convolutional layers). A larger probability $p$ indicates more perturbation in the model weights. $s$ denotes the number of distinct training subsets used for conventional shadow models.

Factor ①: **Consistency with the target model.** Shadow models aim to replicate the behavior of the target model and have similar generalization capabilities [31]. Therefore, we consider the consistency between shadow and target models by their output distributions. We fit training data logits on shadow models to a Gaussian distribution and measure their overlap with the target model using the Bhattacharyya coefficient. Figure 1(b) shows that such consistency is dependent on the specific construction method. Moreover, the level of consistency strongly influences attack performance: the lower the consistency, the weaker the attack performance, e.g., *Conv-0.05* and *Conv-0.08* cases.

Factor ②: **Diversity of shadow models.** We further examine how the diversity of shadow models influences the success of inference attacks on *FC-p*. To validate this, we measure the diversity of shadow models using their output entropy and report the entropy gain from adding more models (starting with 4 shadow models) in Figure 1(c). We find that although more augmented models are used in *FC-p*, their diversity increases less than that of conventional shadow models. In other words,

they offer limited additional information to the subsequent attack model, resulting in only marginal improvements in attack effectiveness.

**Factor ③: Randomness of training subsets.** Since augmented shadow models are statically constructed and share the same training subsets as the original models, we turn to the distribution of these subsets to explain attack effectiveness. For $n$ shadow models, we randomly construct $s$ distinct training subsets from the auxiliary data. When $n > s$, multiple models are trained on the same subset. Figure 1(d) shows that using totally unique training subsets achieves the best attack performance. Distinct training allocations enable shadow models to better capture randomness in data distribution, thereby enhancing inference attack effectiveness.

**Summary:** The above analysis suggests that effective shadow models should capture the randomness inherent in both the training process and the training data. The former necessitates consistency with the target model while ensuring diversity among shadow models, whereas the latter focuses on distinct training subsets. These principles inspire us to design a new shadow model construction algorithm as follows.

## 4 Methodology

Our goal is to improve the efficiency of shadow model construction. As a fundamental component of inference attacks, our threat models are aligned with specific attack settings. Existing inference attacks [1, 9, 14] typically assume that the adversary knows (1) the training algorithm $\mathcal{T}$, e.g., the network architecture and loss function of the target model, enables the training of a similar model from scratch on datasets of the adversary's choice; (2) an auxiliary dataset from the same distribution, which may or may not overlap with the target dataset. Therefore, building on above assumptions, we address the efficiency issue of shadow model construction by proposing a shadow pool training framework, SHAPOOL.

### 4.1 Motivation & Challenges

Shadow models account for a large proportion of the computational overhead in inference attacks, as each model is trained and operates independently, causing the cost to scale significantly with the number of shadow models. To address this issue, we construct shared models that are trained jointly within a single process. The advantage lies in the reuse of sub-networks across models, which significantly enhances construction efficiency.

Specifically, we utilize the MoE mechanism [18, 32, 33] to construct a shadow pool of shared models for inference attacks, as it offers two benefits as follows: (1) *Well-suited structure*: An end-to-end activated pathway in MoE can be equivalent to a shadow model of an identical architecture, which allows us to create a large number of shared models as potential substitutes. Theoretically, an MoE model with $L$ layers, each containing $M$ experts, can form up to $M^L$ unique pathways. (2) *Training efficiency*: The sparse activation in MoE effectively scales model parameters without proportionally increasing computational demands [34]. This design enables the reuse of trained experts across different activated pathways, thereby improving the overall training efficiency of the shared models. However, it is non-trivial to adapt a vanilla MoE to construct the shadow pool due to the following challenges:

*Challenge 1*: **Routing Specialization and Fluctuation.** Existing routing strategies assign each input to the most suitable experts during inference [32, 35], which creates an over-specialized mapping between inputs and pathways. Such specialization not only hinders the ability to capture diverse model behaviors on identical inputs [9], but also results in insufficient randomness of training data distribution across pathways, violating Factor ③. Moreover, the routing fluctuation effect in MoE [36], where the target expert for the same input can change during training, leads to unstable and overlapping usage of training data across pathways.

*Challenge 2*: **Similar Pathways.** Although randomness in the training process, such as weight initialization, mini-batch SGD, and random routing [37], allows variations in expert performance, shared experts across multiple pathways may still lead to similar behaviors, violating Factor ②. In an extreme case, only a pair of experts differ between two highly overlapping pathways. Such similarity leads to redundant and under-informative shared models for subsequent inference attacks.

***Challenge 3*: Generalization Mismatch.** The end-to-end activated pathways exhibit varying degrees of generalization compared to models trained independently, thereby violating Factor ①, as shown in Appendix Figure 5(a). For each pathway in MoE, the training data is typically much smaller compared to that of independently trained models, which leads to some experts being under-trained. On the other hand, the sharing mechanism affects the extent of overfitting in individual pathways.

## 4.2 SHAPOOL: MoE-based Shadow Pool Training Framework

To overcome above challenges, we propose *SHAPOOL*, an MoE-based shadow pool training framework, where activated pathways serve as effective substitutes for independently trained shadow models. As shown in Figure 2(a), it includes three modules, pathway-choice routing, pathway regularization, and pathway alignment, to maintain similar effectiveness and reduce the computational cost of the construction. For clarity, we use the terms 'activated pathway' and 'shared model' interchangeably throughout this paper.

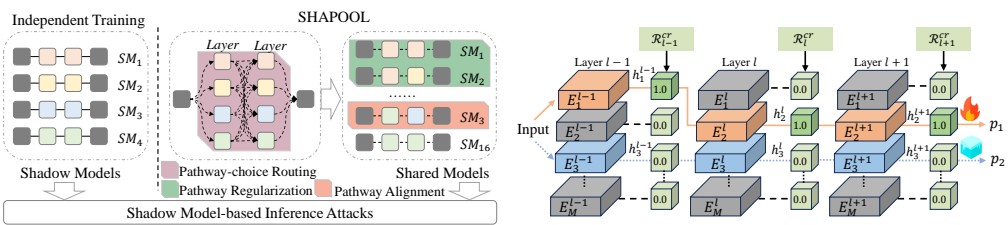

(a) Overview of SHAPOOL.  (b) Illustration of pathway regularization.

Figure 2: (a) Solid lines represent hard connections within shadow models, while dotted lines indicate soft connections across experts in MoE. (b) The sub-networks in orange denote an activated pathway, while those in blue indicate a reference pathway.

**Module 1: Pathway-choice Routing.** To deal with routing specialization and fluctuation, i.e., Challenge 1, we propose a pathway-choice routing strategy, which routes each input to fixed but randomly selected pathways and activates only one expert per layer during training and inference. This design enables us to maintain stable correlations between activated pathways and inputs, as well as to ensure a distinct data allocation for pathway learning.

We implement this pathway-choice routing strategy by establishing a hard mapping between pathways and disjoint training subsets. Given an MoE model with $L$ layers and $M$ experts per layer, we first enumerate all possible pathways as a set $\{\mathbf{P}_w\}_{w=1}^{M^L}$ where $\mathbf{P}_w \subset \mathbb{R}^L$ denotes the indices of activated experts in this pathway. Given a training set $D_{tr}$, we define $\mathbf{B} \subset \mathbb{R}^{|D_{tr}| \times M^L}$ as a binary mapping matrix, where each element $\mathbf{B}_{i,w}$ indicates whether the $w$-th pathway is updated by the input $x_i$ from the training set (1 for update, 0 otherwise). Formally, our pathway-choice router of the $l$-th layer for the $w$-th pathway is defined as

$$\mathcal{R}(x_i; w, l) = \mathbf{B}_{i,w} \cdot \mathbf{1}_{[\mathbf{P}_w[l]=k]}, \tag{2}$$

where $\mathbf{1}_{[\mathbf{P}_w[l]=k]}$ is an $M$-dimensional binary vector indicating which expert is activated at the layer for the $w$-th pathway. Then, the output of the $l$-th expert layer is formulated as:

$$h = \mathcal{R}(x_i; w, l) \cdot \mathbf{E}(x_i). \tag{3}$$

$\mathbf{E}(x_i)$ represents an output matrix of all experts, where each row refers to the output of a single expert for the input $x_i$. We establish a random yet stable assignment between experts and inputs by a mapping matrix $\mathbf{B}$. By controlling the overlap of data points across pathways, we ensure distinct training subsets for shared models and capture randomness in data distribution.

**Module 2: Pathway Regularization.** SHAPOOL leverages pathway regularization to improve the diversity of shared shadow models to address Challenge 2. The core idea is to constrain the outputs of experts and produce diverse model predictions for the same input. We enforce the constraint on activated pathways by introducing two loss terms: similarity regularizer and orthogonal regularizer. Inspired by contrastive learning [38, 39], SHAPOOL activates a pair of pathways in each iteration instead of a single pathway to enhance their differences. During training, we randomly select

an additional pathway as a reference and update the model parameters of the activated pathway. Figure 2(b) illustrates one iteration of the training process for an input.

We first work on the issue of redundant model outputs from different pathways in SHAPOOL. Since model outputs are a key component in various inference attacks, particularly against black-box models [1, 13], we introduce a similarity regularizer to control the alignment of output probability among pathways. Let $p = f(x; P)$ denote the prediction probabilities of an MoE model $f$, where a set of experts along the pathway $P$ are activated. As shown in Figure 2(b), given that two pathways are activated in one iteration, their prediction probabilities are represented as $p_1 = f(x; P_1)$ and $p_2 = f(x; P_2)$, where $P_1 = \{\ldots, E_1^{l-1}, E_2^l, E_2^{l+1}, \ldots\}$ and $P_2 = \{\ldots, E_3^{l-1}, E_3^l, E_3^{l+1}, \ldots\}$ respectively. We use the Kullback–Leibler (KL) divergence to measure the similarity of the output distributions between pairwise pathways [37]. The similarity regularizer $\mathcal{L}_{SR}$ is defined as the negative average of two KL divergence terms calculated from the prediction probabilities, as follows:

$$\mathcal{L}_{SR}(p_1; p_2) = -\frac{1}{2}(KL(p_1||p_2) + KL(p_2||p_1)). \tag{4}$$

We further introduce an orthogonal regularizer to eliminate feature correlation of experts within a single layer. This is important in extreme cases where a pair of activated pathways share all but one expert, leading to highly similar predictions for the same inputs. To address this issue, a strong constraint is imposed from the perspective of expert outputs to enhance their misalignment. In particular, we turn to orthogonal regularization [40, 41, 42] to improve the orthogonality effect in the representation spaces of experts. Rather than using common weight initialization methods, which may exhibit uncertain orthogonality in the convolutional layer [43] and impose overly stringent constraints [40], we apply the regularizer directly to the outputs of the experts and make their representation space as orthogonal as possible. Let $H_1$ and $H_2$ denote two sets of internal activations given a pair of pathways, respectively. We define the orthogonal regularizer $\mathcal{L}_{OR}$ as the sum of the pairwise inner products of the outputs from activated experts across all layers, formally expressed as:

$$\mathcal{L}_{OR}(H_1; H_2) = \sum_{l=1}^{L} \left\| h_i^l \cdot h_j^l(x) \right\|, \tag{5}$$

where $h_i^l$ and $h_j^l$ are the outputs of activated experts in the $l$-th layer for $P_1$ and $P_2$, respectively. Together with the commonly-used cross-entropy loss $\mathcal{L}_{CE}$ for model accuracy objective, the entire training objective of SHAPOOL with respect to a training sample $(x, y)$ is to minimize:

$$\mathcal{L} = \mathcal{L}_{CE}(p_1; y) + \alpha\mathcal{L}_{SR}(p_1; p_2) + \beta\mathcal{L}_{OR}(H_1; H_2). \tag{6}$$

The hyperparameters $\alpha$ and $\beta$ are introduced to balance model utility and the diversity strength of two regularization terms. The training objective in Eq.(6) forces all experts to minimize training accuracy errors while maximizing the diversity of their predictions.

**Module 3: Pathway Alignment.** Challenge 3 points out that shared models in MoE tend to misalign with independent models in terms of generalization. Consequently, this mismatch fails to mimic the behaviors of the target model and degrades the performance of inference attacks. To address this issue, we aim to drive under-trained experts toward the performance of well-trained ones. The core idea is to enhance the memorization capacity of these activated pathways and align their behaviors with those of independent models across different data distributions, through a key module in SHAPOOL termed pathway alignment.

We leverage the commonly used fine-tuning technique to achieve this alignment. A simple approach is to update specific layers of a pre-trained model while keeping the remaining weights frozen [44, 45]. Similarly, we update the model parameters along the activated pathway using a small dataset $D_q$ randomly sampled from $D_{tr}$ (around 10%). Specifically, after Modules 1 and 2, we randomly select $n$ pathways from the pre-trained MoE model $f$ and fine-tune them on $D_q$ by minimizing $\mathcal{L}_{CE}$. We reuse a portion of samples from the overall training set $D_{tr}$ to enrich the data available to each pathway, since each is initially trained on a considerably smaller subset compared to independent shadow models. These selected pathways are further trained on more examples, making their behavior more similar to that of a conventional shadow model and effectively reducing the mismatch with the target model, as illustrated in in Appendix Figure 5(b).

**The Complete Algorithm**: We present the pseudocode for SHAPOOL in Appendix, a specialized MoE model designed for the shadow model construction problem, where activated pathways can replace conventional shadow models in various attacks. Prior to MoE training, we randomly assign

training data to different pathways and record the mapping between each input and its corresponding activated pathway. During training, we promote diversity and informativeness among shared models by regulating pairwise pathway similarity via Eq.(6). Finally, the pathway alignment module post-processes the trained pathways to align their behaviors with those of independently trained models across both training and test data distributions. Last but not least, similar to the shadow model technique, to learn more different aspects in both data distribution and the training process, we can develop multiple independent shadow pools, each providing some shared models for inference attacks.

## 5 Evaluation

This section conducts a comprehensive evaluation of SHAPOOL, demonstrating its effectiveness and efficiency across existing inference attacks under two distinct scenarios: (1) *Resource-Constrained Scenario*: There is a limit to training only a small number of shadow models, such as online inference attacks [21]. In this case, our goal is to enhance the effectiveness of existing attacks while consuming the same or similar amount of computational resources. (2) *Resource-Unconstrained Scenario*: There are no explicit cost constraints, allowing for a large number of shadow models, such as offline data auditing or model vulnerability analysis [22, 23]. We aim to maintain comparable attack performance while significantly improving computational efficiency.

### 5.1 Experiment Settings

**Datasets and Models.** Our evaluation is conducted on three benchmark datasets: CIFAR100 [46], CIFAR10 [46] and CINIC10, and three typical network architectures: ResNet18 [47], VGG16 [48], and WideResNet28-10 [49]. For the configuration of SHAPOOL, we set the number of expert layers $L$ to 4 (3 for WideResNet28-10), the number of experts $M$ to 4, and the number of shared models $n$ to 64. Detailed settings are provided in Appendix C.1.

**Evaluation Protocol and Baseline.** A shadow model-based inference attack pipeline generally consists of three stages: shadow model construction, attack model construction, and inference execution. We consider the conventional shadow model construction approach as the baseline (denoted by BASE), where each shadow model is independently trained on a randomly sampled subset of auxiliary data. Our proposed framework replaces the shadow model construction stage with shared models from the pool, while the other two stages remain unchanged.

**Attack Setup.** Following [1, 9, 16, 50, 51], we assume that an adversary knows the target model's network structure and possesses shadow datasets similar to the target data distribution. We evaluate the effectiveness of SHAPOOL by integrating it into two shadow model-based MIAs: LiRA [9] and RMIA [16], replacing their original shadow model construction. Each method is evaluated in both offline and online modes; the latter generally offers better performance but incurs higher computational cost. Further details are available in Appendix C.2.

**Evaluation Metrics.** Following existing MIAs [9, 29, 52], we adopt two evaluation metrics as performance indicators, i.e., Area Under the ROC Curve (AUC) and True Positive Rate (TPR) at low False Positive Rate (FPR). As for computational costs, we present the wall-clock time (in hours) required for shadow model construction, along with the corresponding percentage reduction in runtime. We report the average results over five independent runs with different random seeds and and release the source code at `https://github.com/BaiLibl/ShadowPool.git`.

### 5.2 Main Results

We evaluate the effectiveness of SHAPOOL in two different scenarios, followed by an investigation into how its several components contribute to the overall attack performance.

**Inference Attacks under Low Computation Budget.** We first analyze the performance of inference attacks under low computational resources, i.e., using a small number of shadow models. To achieve a similar cost, we use one shadow pool for SHAPOOL, while BASE uses four shadow models for LiRA-online and two shadow models for LiRA-offline. SHAPOOL removes pathway regularization and uses only a few fine-tuning epochs when conducting LiRA-offline. Detailed settings are provided in Appendix C.1. Table 1 compares the attack performance of different construction methods.

Table 1: Attack performance and computational cost of LiRA under the constrained resource. We use = to represent changes in the attack performance smaller than 0.02 and TF1 to denote TPR@FPR=1%.

| DATASET | METHOD | RESNET18 | | | VGG16 | | | WIDERESNET28-10 | | |
|---|---|---|---|---|---|---|---|---|---|---|
| | | AUC | TF1 | COST | AUC | TF1 | COST | AUC | TF1 | COST |
| | | | | | LIRA-OFFLINE ATTACK | | | | | |
| CIFAR100 | BASE | 0.73 | 0.22 | 0.5 | 0.69 | 0.16 | 0.4 | 0.76 | 0.30 | 2.4 |
| | SHAPOOL | 0.82 | 0.32 | 0.4 | 0.76 | 0.17 | 0.5 | 0.81 | 0.27 | 1.9 |
| | Δ | +0.09 | +0.10 | - | +0.07 | = | - | +0.05 | -0.03 | - |
| CIFAR10 | BASE | 0.51 | 0.06 | 0.5 | 0.56 | 0.04 | 0.4 | 0.55 | 0.08 | 2.2 |
| | SHAPOOL | 0.63 | 0.09 | 0.5 | 0.64 | 0.09 | 0.4 | 0.60 | 0.08 | 1.6 |
| | Δ | +0.12 | +0.03 | - | +0.08 | +0.05 | - | +0.05 | = | - |
| | | | | | LIRA-ONLINE ATTACK | | | | | |
| CIFAR100 | BASE | 0.90 | 0.34 | 1.0 | 0.86 | 0.25 | 0.9 | 0.90 | 0.36 | 3.6 |
| | SHAPOOL | 0.92 | 0.42 | 1.0 | 0.88 | 0.25 | 0.9 | 0.91 | 0.42 | 3.4 |
| | Δ | +0.02 | +0.08 | - | +0.02 | = | - | = | +0.06 | - |
| CIFAR10 | BASE | 0.69 | 0.11 | 1.0 | 0.67 | 0.07 | 0.9 | 0.67 | 0.09 | 4.4 |
| | SHAPOOL | 0.71 | 0.11 | 1.0 | 0.69 | 0.10 | 1.2 | 0.69 | 0.12 | 4.5 |
| | Δ | +0.02 | = | - | +0.02 | +0.03 | - | +0.02 | +0.03 | - |

Table 2: Attack performance and computational cost of LiRA under the unconstrained resource.

| DATASET | METHOD | RESNET18 | | | VGG16 | | | WIDERESNET28-10 | | |
|---|---|---|---|---|---|---|---|---|---|---|
| | | AUC | TF1 | COST | AUC | TF1 | COST | AUC | TF1 | COST |
| | | | | | LIRA-OFFLINE ATTACK | | | | | |
| CIFAR100 | BASE | 0.81 | 0.36 | 15.4 | 0.72 | 0.25 | 12.1 | 0.77 | 0.37 | 76.8 |
| | SHAPOOL | 0.84 | 0.37 | 1.6 | 0.78 | 0.21 | 2.0 | 0.82 | 0.36 | 7.7 |
| | Δ | +0.03 | = | ↓90% | +0.06 | -0.04 | ↓84% | +0.05 | = | ↓90% |
| CIFAR10 | BASE | 0.57 | 0.11 | 16.8 | 0.54 | 0.06 | 13.7 | 0.56 | 0.09 | 70.8 |
| | SHAPOOL | 0.63 | 0.13 | 1.9 | 0.65 | 0.10 | 1.6 | 0.61 | 0.09 | 6.4 |
| | Δ | +0.06 | +0.02 | ↓89% | +0.11 | +0.03 | ↓88% | +0.05 | = | ↓91% |
| | | | | | LIRA-ONLINE ATTACK | | | | | |
| CIFAR100 | BASE | 0.95 | 0.55 | 30.7 | 0.90 | 0.36 | 24.3 | 0.93 | 0.50 | 153.6 |
| | SHAPOOL | 0.94 | 0.53 | 3.9 | 0.89 | 0.31 | 3.1 | 0.92 | 0.48 | 14.5 |
| | Δ | = | -0.02 | ↓87% | = | -0.05 | ↓88% | = | -0.02 | ↓91% |
| CIFAR10 | BASE | 0.71 | 0.16 | 33.7 | 0.71 | 0.14 | 27.5 | 0.71 | 0.13 | 141.6 |
| | SHAPOOL | 0.70 | 0.14 | 4.0 | 0.69 | 0.13 | 4.8 | 0.70 | 0.13 | 18.1 |
| | Δ | = | -0.02 | ↓88% | -0.02 | = | ↓82% | = | = | ↓87% |

Overall, SHAPOOL demonstrates superior performance in most cases across various settings. For instance, SHAPOOL improves AUC by 5%–12% in LiRA-offline and by around 2% in LiRA-online, respectively. We also observe a relatively modest improvement on the WideResNet28-10 network compared to the other network types, attributed to its fewer expert layers and less diverse shared models. Besides, SHAPOOL shows a more significant improvement in LiRA-offline across evaluation metrics than in LiRA-online. This difference arises because the former attack is less sensitive to the output distribution of shadow models. these results indicate that SHAPOOL enhances inference attack performance by replacing independent shadow models with more shared models under similar computational cost.

**Ultimate Performance of Inference Attacks.** We then evaluate the attack power when a large number of conventional shadow models can be trained. This scenario aims to validate SHAPOOL's training efficiency and ultimate attack performance. Following previous settings [9, 52], we prepare 128 shadow models for LiRA-online (half IN and half OUT) and 64 for LiRA-offline (all OUT). In this case, we construct four shadow pools, each randomly sampling 64 pathways as shared models. Table 2 presents the attack performance and computational cost across different settings. First, for the LiRA-offline attack, our proposed method outperforms the baseline as the size of the shadow pools increases, mostly achieving higher attack performance while significantly reducing the computational cost of conventional shadow model training by an average of 88.6%. Regarding LiRA-online MIA, SHAPOOL achieves comparable performance in terms of AUC, with a slight reduction in TPR@FPR=1%. This is attributed to the high sensitivity and limited robustness of the online mode to the behavior of shadow models on the training data distribution [9]. Overall, our SHAPOOL framework improves the efficiency of shadow model training by approximately

Table 3: Attack performance and computational cost of RMIA.

| Method | RMIA-offline | | | | RMIA-online | | | |
|---|---|---|---|---|---|---|---|---|
| | AUC | TF1 | TF01 | Cost | AUC | TF1 | TF01 | Cost |
| BASE | 0.94 | 0.53 | 0.44 | 7.5 | 0.94 | 0.53 | 0.40 | 16.0 |
| SHAPOOL | 0.93 | 0.52 | 0.42 | 1.8 | 0.93 | 0.53 | 0.39 | 4.0 |
| Δ | = | = | = | ↓76% | = | = | -0.02 | ↓75% |

Table 4: Loss terms.

| $\mathcal{L}_{CE}$ | $\mathcal{L}_{SR}$ | $\mathcal{L}_{OR}$ | AUC | TF1 |
|---|---|---|---|---|
| ✓ | | | 0.83 | 0.35 |
| ✓ | ✓ | | 0.84 | 0.36 |
| ✓ | | ✓ | 0.87 | 0.39 |
| ✓ | ✓ | ✓ | 0.87 | 0.42 |

7.7× compared to BASE on LiRA. Apart from CIFAR10 and CIFAR100, we further validate the performance of SHAPOOL on CINIC10 in Appendix C.3.

**Performance on Different Attacks.** To further validate the board applicability of SHAPOOL across different attacks, we report experimental results on RMIA using ResNet18 and CIFAR100, evaluating AUC, TPR@FPR=1% (TF1), and TPR@FPR=0.1% (TF01) under the unconstrained setting. Specifically, we compare the performance of the original RMIA using 64 independent shadow models with its enhanced version that utilizes shared models from four shadow pools. Table 3 demonstrates that unlike previous methods tailored to specific attack strategies, our approach generalizes across different inference attacks with improved efficiency and adaptability. Moreover, beyond MIAs, we extend our method to PIAs, as detailed in Appendix C.4, further demonstrating its effectiveness and efficiency.

## 5.3 Ablation Study

We investigate how various components in SHAPOOL collectively achieve comparable attack performance using CIFAR100 and ResNet18.

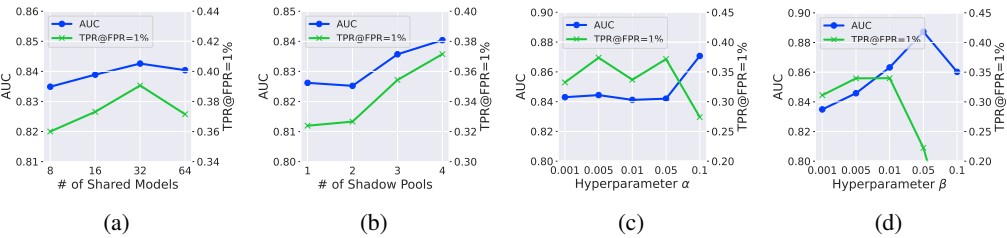

Figure 3: Hyperparameter settings on the LiRA-offline using ResNet18 and CIFAR100.

**Number of Shared Models.** We evaluate the impact of the number of shared models sampled from the shadow pool on attack performance. We test LiRA-offline attacks using four different numbers of shadow models: 8, 16, 32, and 64, using CIFAR100 and ResNet18. Figure 3(a) illustrates a slight performance improvement in both AUC and TPR@FPR=1% as the number of shared models increases; however, this effect diminishes when the number exceeds 32, indicating that while a shadow pool can provide numerous potential shared models, their effectiveness is ultimately limited.

**Number of Shadow Pools.** To approximate the ultimate attack performance, we build multiple shadow pools for shared models in the above experiments. Here, we examine the impact of varying the number of shadow pools on attack performance, ranging from 1 to 4. Figure 3(b) shows that increasing the number of shadow pools improves both evaluation metrics, especially a more noticeable effect on TPR@FPR=1%. This upward trend suggests that SHAPOOL has the potential to further enhance attack performance beyond the results reported in Tables 2 and 3.

**Training Objective.** Path regularization employs a specialized training objective to enhance the diversity of shared models. Here, we examine the relative contributions of its loss terms: $\mathcal{L}_{CE}$, $\mathcal{L}_{SR}$, and $\mathcal{L}_{OR}$. The results in Table 4 indicate that $\mathcal{L}_{OR}$ plays a crucial role in attack performance, whereas adding $\mathcal{L}_{SR}$ results in only a slight improvement in AUC and TPR@FPR=1%, as $\mathcal{L}_{OR}$ imposes a more direct restriction on the experts.

**Regularization Strength.** We further investigate the impact of the regularization strengths $\alpha$ and $\beta$. To control the experiment, we set the other hyperparameter to zero when testing either one. Regarding the hyperparameter $\alpha$, Figure 3(c) demonstrates that the attack performance remains stable as long as $\alpha$ is not very large (e.g., $\alpha \leq 0.05$). And $\beta$ has a similar effect on attack performance in Figure 3(d).

As such, in our experiments, we set it to a value smaller than 0.01. The two regularizers positively influence attack performance when setting their contribution to an appropriate scale. Due to space limitations, additional results under various settings are provided in Appendix C.3.

## 6 Discussion

**Broader Impacts.** This paper advances our understanding of inference attacks in machine learning models. We take a new perspective on investigating the construction of shadow models. By adopting MoE-based shared model training, we not only accelerate the construction but also enhance the effectiveness of these shadow models for inference attacks. Our proposed framework offers model publishers or data owners an efficient tool to audit and assess potential privacy risks when publishing a model or a model service.

**Limitations.** We improve the efficiency of shadow model construction by introducing an MoE-based shadow pool, where activated pathways serve as substitutes for conventional shadow models. However, this approach requires adapting the shadow model architecture to the MoE framework, entailing network architectural modifications. Furthermore, due to the increased number of parameters in MoE, the proposed method is more effective when the adversary possesses a sufficient amount of auxiliary data; its performance may degrade in low-data attack scenarios.

## 7 Conclusion

Targeting the efficiency bottleneck of shadow model construction, a core component of inference attacks, we propose an attack-agnostic solution to address this issue. Instead of independently training shadow models in a conventional manner, we construct a large number of shared models and train them jointly within a single process using the MoE mechanism. While this reuse of sub-networks improves the construction efficiency, dependent shadow models may negatively impact the attack performance. We propose three enhancement modules to address these issues and enable shared models from the pool to serve as effective substitutes. Our approach replaces conventional shadow models with shared models, significantly reducing construction costs while preserving competitive attack performance across various datasets. Moreover, it can be seamlessly integrated into a wide range of inference attacks. Future work can explore refined parameter-sharing strategies to further balance attack efficiency and effectiveness. Additionally, adaptive shadow model training mechanisms remain an interesting direction.

## Acknowledgments

This work was supported by the National Natural Science Foundation of China (Grant No: 92270123 and 62372122), the Research Grants Council (Grant No: 15208923, C2004-21GF, C2003-23Y), and the PolyU Research Centre for Privacy and Security Technologies in Future Smart Systems, Hong Kong SAR, China.

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

# Appendices

---

**Algorithm 1:** Shadow Model-based Inference Attacks

---

**Input:** Training algorithm $\mathcal{T}$, Target model $M_T$, Auxiliary dataset $D_A$, Query examples $Q$, Attack algorithm $\mathcal{T}'$.
**Output:** Inferred secret $s'$.
    // Shadow Model Construction
1  Construct shadow training subsets $D_{A_1}, D_{A_2}, ..., D_{A_M}$ randomly sample from $D_A$ ;
2  **for** $i = 1; i \leq M$ **do**
3    |   Train a shadow model $M_{S_i} \leftarrow \mathcal{T}(D_{A_i})$;
4  **end**
    // Attack Model Construction
5  $D_{att} \leftarrow \emptyset$;
6  **for** $i = 1; i \leq M$ **do**
7    |   $(D_{A_i}, s_i) \leftarrow M_{S_i}(D_{A_i})$ ;       // $s_i$: ground-truth secret, e.g., membership status
8    |   $D_{att} \leftarrow D_{att} \cup (D_{A_i}, s_i)$;
9  **end**
10 Train an attack model $\mathcal{A} \leftarrow \mathcal{T}'(D_{att})$;
    // Inference Attack
11 $s' \leftarrow \mathcal{A}(M_T(Q))$;
12 **return** $s'$;

---

## A   Related Work

### A.1   Shadow Model-based Inference Attacks

The shadow model training technique aims to mimic the behavior of the target model by preparing shadow models with the same network architecture on similar datasets, which provides the ground truth of inference results and outputs for the meta-classifiers used in inference attacks [1, 8, 53, 13, 14], as outlined in Algorithm 1. Several popular inference attacks highlight the versatility of this technique. For instance, shadow model-based membership inference attacks leverage shadow models to construct binary inference classifiers or hypothesis tests [1, 9]. Similarly, property inference attacks [13, 14, 54] use shadow models to build meta-classifiers that reveal sensitive property information in property inference attacks. Moreover, in advanced model inversion attacks [55], shadow model training facilitates the joint supervised training of the encoder and decoder with the collection of shadow-target model pairs.

Current research on shadow models mainly examines their training setup in relation to the target model, focusing on factors such as employing an identical network architecture [56, 10] and analyzing the effects of varying shadow training data distributions [15, 10, 9]. Departing from this focus, we aim to explore advanced methods for shadow model construction and enhance the attack efficiency of shadow model-based inference attacks.

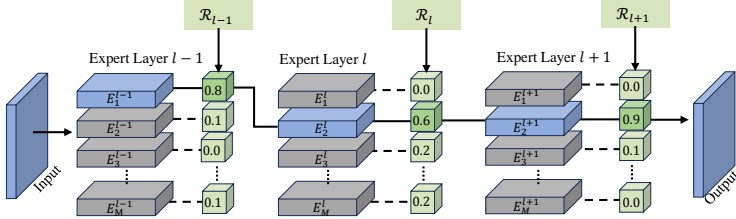

Figure 4: Example of a vallina MoE. For simplicity, non-expert layers are omitted here. An activated end-to-end pathway, highlighted in blue, is formed by the experts selected through the input-dependent routing function $\mathcal{R}$, along with the fixed non-expert layers.

## A.2 Mixture of Experts

As a special instance of conditional computation [19, 20], sparsely-gated MoE [18, 33, 32, 35, 57, 58, 59] contains a group of identical sub-networks (i.e., *experts*) and makes input-dependent predictions according to the choices of experts. It increases model parameters while maintaining a similar computational cost per example during inference, powering kinds of AI domains, including multi-task learning [18, 33], NLP [32, 35], and computer vision [57, 58, 59]. For simplicity, we refer to it as MoE moving forward.

Let first introduce a vanilla MoE model consisting of $L$ expert layers, each containing $M$ experts with identical network structures and a router $\mathcal{R}$, as illustrated in Figure 4. Given an input $x$, the output of an expert layer is computed as the summation of the outputs from the top $K$ experts selected by $\mathcal{R}$:

$$h = \sum_{k=1}^{K} \mathcal{R}(x) \cdot E_k(x), \ \mathcal{R}(x) = \text{TopK}(\mathcal{G}(x), K), \tag{7}$$

where $E$ is a learnable expert (i.e., sub-network), $\mathcal{G}$ is a routing strategy based on trainable networks or heuristic functions and $\text{TopK}(\cdot, K)$ refers to selecting the largest $K$ values with the remaining values set to zero. Figure 4 illustrates an iteration of the MoE training process.

Key research on MoE models focuses on effective routing strategies that navigate inputs to the most suitable experts and achieve expert load balance [60, 61, 62, 20]. The mainstream approach, known as the input-choice routing strategy, assigns each input to one or more experts. For example, Switch Transformers [62] uses a softmax layer to select the top expert for each example. Similarly, Base Layers [61] employ a linear assignment function to allocate tokens to experts. Hash Layer [63] utilizes hashing to assign input tokens to experts, whereas THOR [37] activates experts randomly for each input. Alternatively, rather than selecting one or two top-scoring experts for each input, [60] allows each expert to pick the top-k inputs, ensuring perfect load balancing. Unlike previous works focused on the MoE mechanism itself, our goal is to extend it to shadow model construction and reduce the high computational cost of inference attacks.

# B  Case Study: Model Augmentation-based Shadow Models

Model augmentation introduces variations in data, architectures, or training procedures to enhance machine learning models, improving generalization and robustness. Techniques such as dropout and data augmentation have been widely used for ensemble learning [25], continual learning [26], adversarial attacks [27, 28, 29], and so on. Neural masking [30, 28] is a typical model augmentation approach, which randomly prunes the weight of model parameters of a base model $W_f$ with a probability of $p$. The resulting weights of an augmented model $W_{aug}$ are computed as follows:

$$W_{aug} = W_f \odot (\mathbf{1} - \xi(W_f, p)) \tag{8}$$

where $\mathbf{1}$ denotes the all-ones matrix with the same size as the weight of the base model, $\odot$ represents the Hadamard product that performs element-wise multiplication on the weights, and $\xi$ refers to a specific augmentation technique.

Model augmentation provides an alternative approach to generating multiple augmented models derived from a base network. These augmented models are post-processed variations of the base network, which are not stored or trained, thus reducing the overall cost of shadow model construction. Specifically, we employ neural masking to enlarge the set of trained shadow models for inference attacks by scaling model weights and generating diverse augmented models.

We explore varying levels of weight modification by focusing on either the fully connected layers (i.e., *FC-p*) or the convolutional layers (i.e., *Conv-p*). We target the last fully connected layers, closest to the model outputs and are commonly utilized in various inference attacks [9, 13, 14, 31, 64]. Conversely, pruning weights in the convolutional layers proves advantageous due to its impact on a larger number of weights, resulting in more diverse augmented models. For example, in ResNet18, the last fully connected layer accounts for only 0.46% of the parameters, while the convolutional layers comprise approximately 99%.

We evaluate different augmentation settings on an MIA, LiRA [9], using CIFAR100 and ResNet18. To ensure the augmented models remain close to the base network while preserving similar generative capacity, we carefully adjust the scale of weight modifications, constraining the test error loss to

within 10%. For this, we set thresholds of 0.1 and 0.2 for *FC* and 0.05 and 0.08 for *Conv*. We start by constructing base shadow models in quantities of 4, 8, 16, 32, and 64, then generate one augmented model for each base model, as outlined in Algorithm 2. As a result, the total number of shadow models doubles after augmentation for *Conv-p* and *FC-p*.

---

**Algorithm 2:** Model Augmentation-based Shadow Model Construction

**Input:** Training algorithm $\mathcal{T}$, Auxiliary dataset $D_A$, Augmentation algorithm $Aug$.
**Output:** Shadow models $\{M_{S_i}, M'_{S_i}\}_{i=1}^{M}$.

1 Construct shadow training subsets $D_{A_1}, D_{A_2}, ..., D_{A_M}$ randomly sample from $D_A$ ;
2 **for** $i = 1; i \leq M$ **do**
3     Train base model $M_{S_i} \leftarrow \mathcal{T}(D_{A_i})$;
4     Augment model $M'_{S_i} \leftarrow Aug(M_{S_i})$;
5 **end**
6 **return** $\{M_{S_i}, M'_{S_i}\}_{i=1}^{M}$;

---

**Algorithm 3:** SHAPOOL: MoE-based Shadow Model Construction

**Input:** MoE model $f$, training dataset $D_{tr}$, fine-tuning dataset $D_q$, number of experts $M$, number of layers $L$, number of shared models $n$.
**Output:** Trained MoE model $f$, data-to-pathway mapping matrix $\mathbf{B}$, selected pathway set $S$.
   // Before training: Pathway-choice routing
1 Randomly split $D_{tr}$ into $M^L$ subsets: $D_{tr}^s$;
2 Enumerate all possible pathways $\{\mathbf{P}_w\}_{w=1}^{M^L}$;
3 Construct mapping matrix $\mathbf{B}$ between $D_{tr}^s$ and $\{\mathbf{P}_w\}_{w=1}^{M^L}$;
   // In training: Pathway regularization
4 **foreach** $(x, y) \in D_{tr}$ **do**
5     Randomly select two different pathways $P_1$, $P_2$;
6     Compute intermediate output $h_1$ using Eq. (3);
7     Compute prediction $p_1 \leftarrow f(x; P_1)$;
8     Compute intermediate output $h_2$ using Eq. (3);
9     Compute prediction $p_2 \leftarrow f(x; P_2)$;
10     Update model parameters of $f$ using Eq. (6);
11 **end**
   // Fine tuning: Pathway alignment
12 Randomly select $n$ pre-trained pathways $S$;
13 **for** $i = 1; i \leq |S|$ **do**
14     **foreach** $(x, y) \in D_q$ **do**
15        Compute prediction $p \leftarrow f(x; S_i)$;
16        Update model parameters of $f$ using $\mathcal{L}_{CE}(p; y)$;
17     **end**
18 **end**
19 **return** $f$, $\mathbf{B}$, $S$;

---

## C  Experimental Details and Additional Results

### C.1  Model Setup

In our experiments, both the target and conventional shadow models adopt the same architecture. Excluding the initial and classification layers, we construct the shadow pool by dividing the remaining architecture into multiple expert layers. They are trained for 100 epochs with a batch size of 64 and an initial learning rate of 0.1. We use the SGD optimizer with a weight decay of $5 \times 10^{-4}$ and a momentum of 0.9, along with a cosine learning rate schedule [65] for optimization.

Our experiments replace the conventional shadow models with shared models generated from SHAPOOL. For LiRA-offline, a relatively weaker attack, we remove the pathway regularization (PR) module from SHAPOOL and set the fine-tuning epoch to 3 in the pathway alignment (PA) module. In contrast, LiRA-online is equipped with the PR module for SHAPOOL and sets the fine-tuning epoch to 10 in PA. This difference arises from the sensitivity of attacks for shadow models.

All experiments are implemented in Pytorch and performed on an NVIDIA RTX-3090 server with the Ubuntu operating system.

## C.2 Off-the-shelf MIAs

We adopt two off-the-shelf MIAs to demonstrate the effectiveness of SHAPOOL, including:

**LiRA** [9] requires training multiple shadow models, where half are IN models trained on the query example, and the remaining half are OUT models that have not seen the example. It collects scaled logits from shadow models, fits them into two Gaussian distributions, and uses a likelihood-ratio test to determine the membership status of the query example. There are two attack modes: LiRA-online, which relies on two Gaussian distributions for members and non-members, and LiRA-offline, which relies solely on the Gaussian distribution of non-members.

**RMIA** [16] introduces a fine-grained modeling of the null hypothesis (OUT models) within the likelihood ratio test framework. As with LiRA, it comes in two variants: RMIA-online and RMIA-offline. By introducing reference models and population data samples to reduce reliance on shadow models, both modes achieve comparable performance.

In our experiments, we report the average attack results of 1,000 query examples.

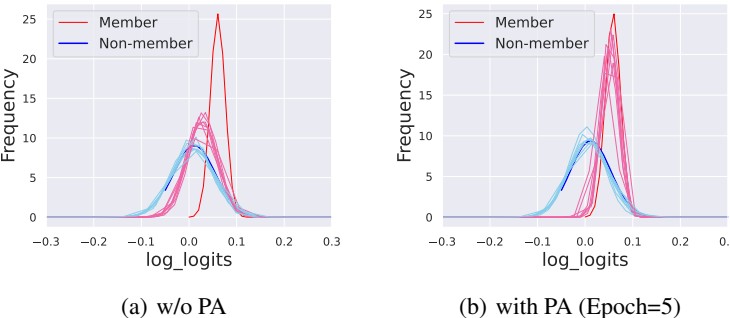

(a) w/o PA  (b) with PA (Epoch=5)

Figure 5: (a) Distribution of shared models of a vanilla MoE. (b) Distribution of shared models after pathway alignment. We present the model output (i.e., scaled logits) distributions for both training (member) and test (non-member) data points using ResNet18 and CIFAR100. The red and blue curves represent the distributions of the target model, while the pink and sky-blue curves illustrate that of shared models from a shadow pool. Since the target model is trained independently, we align the outputs of shared models to better resemble it through pathway alignment.

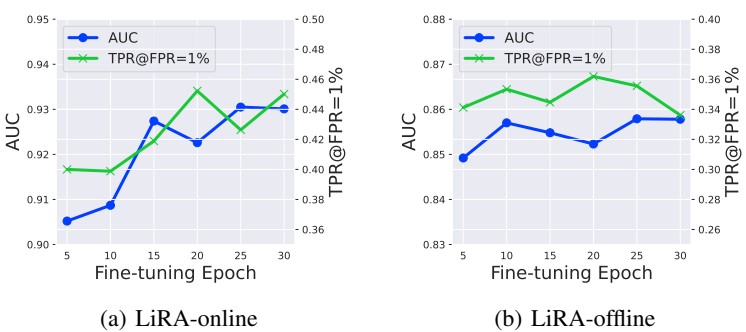

(a) LiRA-online  (b) LiRA-offline

Figure 6: Effect of Fine-tuning Epoch.

## C.3 Additional Experimental Results

### C.3.1 Comparison with Other Baselines

We consider the BASE method as an ideal baseline for training shadow models, as it independently trains shadow models with the same architecture and same data distribution as the target. As such, BASE is a natural and widely adopted approach, especially in black-box MIAs. The only disadvantage is its high computational cost, which is the main motivation of this paper to train shadow models with lower costs.

Regarding alternative shadow model construction strategies, naive approaches such as simple model augmentation have demonstrated only limited effectiveness. In addition, R2 [66] introduces attack D, which leverages distilled models to approximate the OUT world of a query example. We refer to this baseline as *Distilled*. As shown in Table 5, this approach delivers suboptimal performance regardless of whether 64 or 128 distilled models are employed. Such results can be attributed to two reasons. First, while distilled models focus on the OUT world, SHAPOOL considers both IN and OUT worlds. Moreover, distilled models are not well-suited for constructing IN worlds, as the soft-labeling technique mitigates model overfitting—a common defense against MIAs [67, 68]. Second, each distilled model is still trained from scratch using soft-labeled population samples, meaning it does not reduce the training cost of shadow models. In contrast, SHAPOOL refines the shadow model training process by utilizing shared models, significantly reducing the computational cost of MIAs.

Table 5: Various baselines on CIFAR100 and ResNet18 (LiRA-online).

| Method | AUC | TF1 | Cost(h) |
|---|---|---|---|
| BASE | 0.94 | 0.53 | 16.0 |
| Distilled(64) | 0.85 | 0.17 | 1.85 |
| Distilled(128) | 0.83 | 0.21 | 3.45 |
| SHAPOOL | 0.93 | 0.53 | 4.0 |

### C.3.2 Performance on More Datasets

**Attack Performance and Computational Cost on CINIC10.** We now evaluate the performance of SHAPOOL on the CINIC10 dataset using ResNet18. The experimental results in Table 6 demonstrate that our proposed method significantly reduces training costs while preserving comparable attack effectiveness, thereby enabling low-cost yet high-accuracy inference attacks. Moreover, Figure 7 presents the ROC curves of our proposed approach on CINIC10, further demonstrating the superiority of SHAPOOL on attack performance.

Table 6: Attack performance and computational cost of LiRA on CINIC10. TF1 denotes TPR@FPR=1% and results = indicate changes in attack performance smaller than 0.02.

| METHOD | LIRA-OFFLINE | | | LIRA-ONLINE | | |
|---|---|---|---|---|---|---|
| | AUC | TF1 | COST | AUC | TF1 | COST |
| | CONSTRAINED SCENARIO | | | | | |
| BASE | 0.61 | 0.11 | 1.25 | 0.59 | 0.09 | 1.25 |
| SHAPOOL | 0.68 | 0.13 | 1.24 | 0.75 | 0.15 | 1.55 |
| Δ | +0.07 | +0.02 | - | +0.16 | +0.06 | - |
| | UNCONSTRAINED SCENARIO | | | | | |
| BASE | 0.61 | 0.14 | 40.3 | 0.77 | 0.22 | 82.8 |
| SHAPOOL | 0.70 | 0.15 | 4.8 | 0.77 | 0.21 | 6.2 |
| Δ | +0.09 | = | ↓88% | = | = | ↓93% |

**Attack Performance and Computational Cost on TinyImageNet.** In addition to the relatively small-scale datasets used in the previous experiments, we further evaluate SHAPOOL on a larger and more challenging dataset. Specifically, we conduct experiments on TinyImageNet, a subset derived from ImageNet, using ResNet18 under the unconstrained setting. Table 7 provides further evidence supporting the effectiveness of our approach.

Table 7: Performance and cost of RMIA on TinyImageNet.

| Method | RMIA-OFFLINE | | | | RMIA-ONLINE | | | |
|---|---|---|---|---|---|---|---|---|
| | AUC | TF1 | TF01 | COST | AUC | TF1 | TF01 | COST |
| BASE | 0.99 | 0.95 | 0.86 | 28.1h | 0.99 | 0.94 | 0.84 | 61.9h |
| SHAPOOL | 0.99 | 0.94 | 0.87 | 7.9h | 0.99 | 0.95 | 0.88 | 16.1h |
| Δ | = | = | = | ↓72% | = | = | +0.04 | ↓74% |

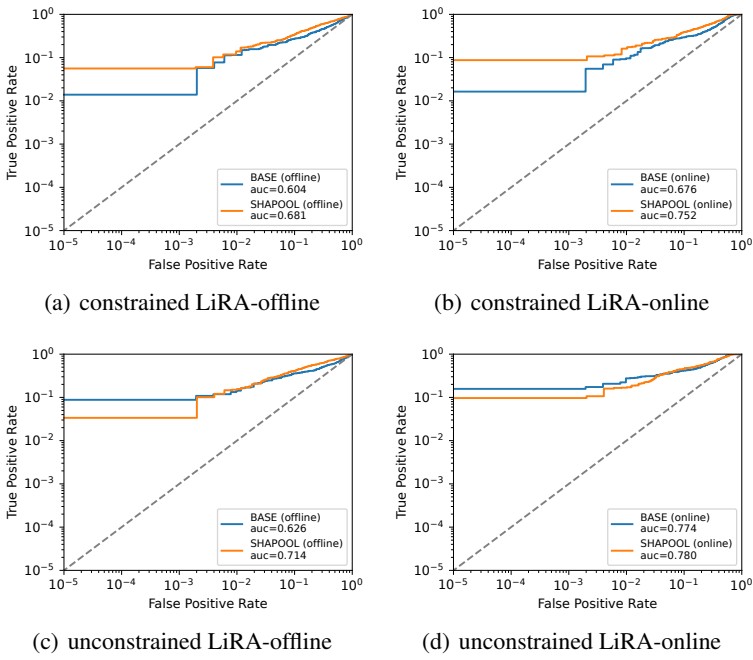

(a) constrained LiRA-offline       (b) constrained LiRA-online

(c) unconstrained LiRA-offline      (d) unconstrained LiRA-online

Figure 7: Comparing the true positive rate vs. false positive rate of our method on the CINIC10 dataset. (a)-(b) constrained scenarios for LiRA-offline/online. (c)-(d) unconstrained scenarios for LiRA-offline/online.

### C.3.3 More Ablation Studies

We conduct additional ablation studies to investigate the impact of different settings on attack performance using CIFAR100 and ResNet18, where a shadow pool is prepared and shared models are randomly selected accordingly.

**Size of Training Set** $|D_{tr}|$**.** SHAPOOL constructs a shadow pool with numerous shared models, resulting in more parameters compared to a single shadow model, which naturally demands more training data. We now examine the impact of training set size on attack performance, relative to the size of the training subset used for conventional shadow model training. Table 8 presents the results for different scales of training datasets. $|D_{A_0}|$ represents the size of the training subset for a shadow model trained independently. Both $|D_{tr}|$ and $|D_{A_0}|$ are sampled from the auxiliary dataset, but we use different notations to distinguish the training set of the shadow pool from that of an individual shadow model. We observe that the performance of SHAPOOL improves significantly after incorporating an additional 20% of data points, and then stabilizes with further increases.

**Size of Fine-tuning Set** $|D_q|$**.** Pathway alignment fine-tunes the selected pathways to reduce generalization mismatch on the training set $D_q$. Here, we examine the effect of the fine-tuning training set size, as shown in Table 10. The percentage represents the ratio of $D_q$ to a training set $D_{tr}$. Notably, there is an overlap between $D_q$ and $D_{tr}$, indicating that the training set can be reused without introducing new data. We find that even a small fine-tuning dataset can achieve stable attack performance.

**Fine-tuning Epochs.** We explore the effect of the fine-tuning epochs in pathway alignment, as shown in Figure 6. The results show that the fine-tuning epoch has a limited effect on the LiRA-offline attack, while a larger epoch (e.g., greater than 15) leads to nearly stable attack performance for both attacks. This difference stems from specific algorithms: LiRA-offline relies on non-member output distributions, which are initially similar in Figure 5. This demonstrates that our pathway alignment requires minimal fine-tuning costs to mitigate generalization mismatch.

**Number of Experts.** The above experiments use four experts in SHAPOOL for ResNet18 by default; now, we explore whether other settings influence the attack performance. As shown in Table 9, we

find that introducing more experts improves attack performance, especially for the LiRA-online attack, as more experts enhance the diversity among shared models.

Table 8: Effect of the training set $|D_{tr}|$ on TF1.

| $|D_{tr}|/|D_{A_0}|$ | LiRA-online | LiRA-offline |
|---|---|---|
| 1.0 | 0.33 | 0.22 |
| 1.2 | 0.37 | 0.28 |
| 1.4 | 0.38 | 0.30 |
| 1.6 | 0.36 | 0.32 |

Table 9: Effect of the number of experts.

| $M$ | LiRA-online | LiRA-offline |
|---|---|---|
| 3 | 0.39 | 0.35 |
| 4 | 0.42 | 0.33 |
| 5 | 0.44 | 0.32 |
| 6 | 0.47 | 0.28 |

Table 10: Effect of the fine-tuning set $|D_q|$ on TF1.

| $|D_q|$ | LiRA-online | LiRA-offline |
|---|---|---|
| 1000 (2.5%) | 0.28 | 0.29 |
| 3000 (7.5%) | 0.41 | 0.31 |
| 5000 (12.5%) | 0.40 | 0.29 |

**Performance under Fixed Costs.** Take LiRA-online on CIFAR100 with ResNet18 as an example. A single SHAPOOL outperforms four individual shadow models *under the same compute budget*. Furthermore, four SHAPOOLs—at a cost equivalent to 16 individual models—nearly match the performance achieved by using 128 shadow models. Table 11 suggests that SHAPOOL consistently outperforms LiRA across a range of fixed compute budgets.

Table 11: Performance of LiRA-online with varying budgets on CIFAR100 (AUC/TF1)

| #SHAPOOL (=#SM) | 1 (=4) | 2 (=8) | 3 (=12) | 4 (=16) |
|---|---|---|---|---|
| BASE | 0.90/0.34 | 0.93/0.44 | 0.94/0.46 | 0.93/0.49 |
| SHAPOOL | 0.92/0.42 | 0.93/0.48 | 0.93/0.50 | 0.94/0.53 |

---

**Algorithm 4:** SHAPOOL-based Property Inference Attacks

**Input:** Predefined values $t_0$ and $t_1$; Attack dataset $D_{attack}$; Shadow dataset $\mathcal{D}$; Attack algorithm $\mathcal{A}$; Target model $M$; MoE models $f_0$ and $f_1$; Epoch $E$

1 ; **Output:** Inferred property value $\hat{t} \in \{t_0, t_1\}$
2 **for** $i \in \{0, 1\}$ **do**
3     Initialize mapping matrix of $f_i$ via pathway-choice routing;
4     **for** $j = 1$ **to** $E$ **do**
5         Sample $D_{i,j}^1, D_{i,j}^2 \subset \mathcal{D}$ such that $P(D_{i,j}^1) = P(D_{i,j}^2) = t_i$;
6         Randomly select pathways $p_1$ and $p_2$ in $f_i$;
7         Train $p_1$ and $p_2$ jointly using pathway regularization on $D_{i,j}^1$ and $D_{i,j}^2$;
8     **end**
9     Randomly select a set of pathways $S \subset f_i$;
10     **foreach** *pathway* $p \in S$ **do**
11         Record confidence scores: $c_i \leftarrow f_i(D_{attack}; p)$;
12     **end**
13 **end**
14 Construct $\mathcal{A}$ using distributions: $N_0 \sim c_0, \; N_1 \sim c_1$;
15 Collect confidence scores: $\mathbf{c} \leftarrow M(D_{attack})$;
16 Compute probability: $\ell_0 \leftarrow \mathcal{A}(\mathbf{c} \mid N_0), \; \ell_1 \leftarrow \mathcal{A}(\mathbf{c} \mid N_1)$;
17 **return** $\hat{t} \leftarrow \arg \max_{i \in \{0,1\}} \ell_i$;

---

### C.4 Extension to Property Inference Attacks

Although our experiments primarily focus on MIAs, SHAPOOL can be extended to other attacks such as black-box property inference attacks (PIAs), which aim to distinguish statistical properties (e.g., between $t_0$ and $t_1$) [13, 14]. To adapt SHAPOOL for PIAs, we can modify the training data of each pathway so that a SHAPOOL is trained on data satisfying $t_0$ (or $t_1$), thereby replacing the multiple shadow models traditionally trained for each predefined property, as illustrated in Algorithm 4.

Table 12: Performance of BASE and SHAPOOL on PIAs using the Census dataset (Accuracy)

| | Acc (Property=sex) | Acc (Property=race) | Cost (h) |
|---|---|---|---|
| BASE | 0.5125 | 0.8594 | 12.8 |
| SHAPOOL | 0.5243 | 0.8438 | 0.9 |
| $\Delta$ | = | = | ↓92% |

Following prior works on PIAs [14], we conduct experiments on the Census dataset [69], focusing on two target properties: sex ("Female") and race ("Black"). We aim to distinguish between 0.3 and 0.5 for the sex property, and between 0.05 and 0.15 for the race property. Specifically, we use a four-layer MLP with layer sizes of 32, 16, 8, and 4, employing ReLU activation functions. For the baseline PIAs (i.e., BASE), we build 32 shadow models for each predefined ratio, whereas in our method (i.e., SHAPOOL), we construct four shadow pools, each consisting of four MoE layers with two experts per layer, and select eight pathways for inference attacks. The comparison results, presented in Table 12, demonstrate that our method achieves comparable inference performance while significantly reducing training costs.

