# OpenReview forum: "Toward Efficient Inference Attacks: Shadow Model Sharing via Mixture-of-Experts"
_NeurIPS.cc/2025/Conference — NeurIPS 2025 poster_

### Official Review · Reviewer_xfRo · 2025-06-06

**Clarity:** 3
**Significance:** 3
**Originality:** 3
**Rating:** 4
**Confidence:** 4

**Summary:**

The paper addresses the computational inefficiency in shadow model construction for inference attacks (e.g., membership inference) against machine learning models. The authors propose SHAPOOL, a framework leveraging the Mixture-of-Experts (MoE) mechanism to train multiple “shared” shadow models within a single process. Experiments across multiple datasets and settings demonstrate a significant reduction in computational costs, as well as competitive attack performance, compared to traditional approaches.

**Questions:**

- I'm a bit confused by the description of the method in Module 3: Pathway Alignment. The aim is to better mimic the behaviors of the target model. Does this process require access to confidence scores from the target model, align results only between experts within the MoE, or simply fine-tune certain experts using additional subsets of the training data?

- The introduction discusses three existing methods for efficient shadow model training: QMIA, RMIA, and SNAP. But the experimental section only provides a direct comparison with RMIA. Is there any reason why QMIA and SNAP were not included in the evaluation?

- I understand that the authors focus on three specific factors in Section 3 to maintain consistency with the subsequent challenges and method design. I wonder if there were additional aspects that could impact MIA performance. Exploring or discussing the potential impact of these additional factors would be more intriguing.

- It's better to provide ROC curves for MIA result comparisons.

**Ethical Concerns:**

["NO or VERY MINOR ethics concerns only"]

**Final Justification:**

The authors’ response has generally addressed my concerns. I would like to maintain my original assessment, as I believe there is still room for improvement in the manuscript.

**Limitations:**

yes

**Paper Formatting Concerns:**

Not found

**Quality:**

3

**Strengths And Weaknesses:**

### Strengths
- The proposed issue, computational cost of training shadow models, is a pressing bottleneck in the privacy attack community, especially for attacks such as LiRA.
- Adapting MoE to address the issue is novel.
- Comprehensive experiments across multiple settings demonstrate the effectiveness of the proposed method.


### Weaknesses
- Some parts seem confusing to follow.
- Experiments could be further enhanced.

---

> ### Author Rebuttal · Authors · 2025-07-31
>
> We thank the reviewer for the constructive comments and address each issue raised below in detail.
>
> > Q1: Clarification about pathway alignment.
>
> The pathway alignment module fine-tunes selected experts using additional subsets of the training data to mitigate under-training issues in the MoE model.
> By “mimic the behavior,” we mean training in a similar manner as the target model. But using confidence scores from the target model would result in a different training paradigm and is shown to reduce MIA effectiveness, as discovered in knowledge distillation-based defenses [1, 2].
> [1] Tang X, Mahloujifar S, Song L, et al. Mitigating membership inference attacks by {Self-Distillation} through a novel ensemble architecture[C]//31st USENIX security symposium (USENIX security 22). 2022: 1433-1450.
> [2] Mazzone F, Van Den Heuvel L, Huber M, et al. Repeated knowledge distillation with confidence masking to mitigate membership inference attacks[C]//Proceedings of the 15th ACM Workshop on Artificial Intelligence and Security. 2022: 13-24.
>
> > Q2: About baseline choices.
>
> QMIA and SNAP are specific inference attack methods rather than shadow model construction techniques. QMIA does not use shadow models, making it incompatible with our framework, while SNAP focuses on property inference via poisoning, which falls outside the scope of our MIA experiments. In contrast, RMIA relies on shadow models, so we include them for comparison to evaluate how SHAPOOL enhances efficiency by replacing the original shadow model construction—i.e., comparing RMIA+BASE with RMIA+SHAPOOL.
>
> > Q3: Additional factors for MIA performance.
>
> We acknowledge that MIA performance can be influenced by additional factors beyond the three key ones we focus on. In particular, while this work focuses on MIA attacks from a black-box perspective, in a white-box setting where the target model’s parameters are accessible, alignment between the shadow models and the target model in parameter space, such as producing similar gradients or intermediate results, may also influence attack effectiveness. We will consider these factors when we extend this work to white-box settings.
>
> > Q4: Suggestions about ROC curves.
>
> We agree that ROC curves offer valuable insights. Due to format constraints in this rebuttal, we are unable to enclose the full ROC curves here but will provide them in the final version of this paper.

---

> > ### Comment · Reviewer_xfRo · 2025-08-04
> >
> > Thanks for the response. It generally resolves my concerns.

---

> > > ### Author Response · Authors · 2025-08-06
> > >
> > > We appreciate you taking the time to review our work and provide helpful comments. Your feedback means a lot to us. We truly appreciate the improved score. Thank you once again!

---

### Official Review · Reviewer_rfXF · 2025-06-12

**Clarity:** 4
**Significance:** 3
**Originality:** 3
**Rating:** 5
**Confidence:** 5

**Summary:**

The paper introduces a novel method of training shadow models for inference attacks that is more computationally efficient and in specific settings outperforms the baseline method of training shadow models independently.

**Questions:**

1. Could you explain what "Extension to Other Inference Attacks" means and what you mean by prior work is "coupled with specific algorithms"?

2. How does the full ROC-curve comparison between your attack and LiRA & RMIA look like for the experiments you have already run?

3. Does your attack perform as well as RMIA in the low reference model (<= 4) setting?

4. Does your attack outperform (in terms of computational time and attack performance) LiRA across a range of fixed compute budgets?

**Ethical Concerns:**

["NO or VERY MINOR ethics concerns only"]

**Final Justification:**

The authors have taken the time to faithfully engage in the rebuttal process and have addressed all of the concerns that I have raised, even responding with updated results. Therefore I feel confident in recommending this paper to be accepted. I would just like to point out that there are two main results from the discussions that I would like to see in the main paper as well, which I had also mentioned to the authors during the rebuttal discussion. Namely:

1. Comparison with RMIA in constrained resource setting.
2. Comparison between SHAPOOL and LiRA/RMIA under fixed compute budget

**Limitations:**

yes

**Quality:**

3

**Strengths And Weaknesses:**

**Strengths**
1. Their method is novel and interesting.
2. The improvements to computational cost appear significant.

**Weaknesses**
_1. Unsubstantiated Claims_
There are some claims that are left entirely unsubstantiated in the paper. Specifically, the authors make the claim that their method is "more generalizable" than prior work. However, I do not fully understand the claim that prior work is "coupled with specific algorithms" and is not cited in the Introduction. What do the authors mean by this and where does this claim come from? Furthermore in the Evaluation section under "Extension to Other Inference Attacks", the authors claim that Table 3 shows that their method generalizes across different attacks. But I do not see how Table 3 shows that since the attack they compare against (RMIA) is also a membership inference attack. I was expecting the authors to compare their SHAPOOL method with LiRA and RMIA across different attack types, e.g., property inference and model inversion in order to make this claim.

_2. Additional Experiments_
I am not entirely convinced that the authors' main claims are fully substantiated either. First, Tables 1 and 2 are missing TPR@FPR=0.1%, even though the authors had indicated this was a metric they were interested in studying. Furthermore, I cannot find full ROC curves for any of the experiments and this is not discussed in the experiments. Specifically, I would like to see whether there are regions of FPRs that their method underperforms prior work. Lastly, Tables 1 and 2 only really compare one configuration for the constrained and un-constrained settings, each. I do not think this is enough for the authors to claim that the computational cost gains are always justified. Here, I think it's important for the authors to vary the number of shadow models and shadow pools for LiRA and SHAPOOL, respectively and compare the two based on the computational cost. Basically, the analogue of Figure 3 from the arXiv version of the RMIA paper.

_3. Unfair comparisons_
I am not really sure why the authors are comparing their method against LiRA under the constrained scenario. Shouldn't they compare against RMIA since it is already known that RMIA beats LiRA in resource-constrained settings? Also it seems odd that Table 3 uses 64 shadow models for RMIA when 4/8 is supposedly enough. Again going back to point 2. this I believe further motivates the authors to do a comparison between SHAPOOL, LiRA, and RMIA across different number of models/computational costs.

_4. Writing Improvements_
Just some minor points, "Bhattacharya coefficient", "CINC10" are missing citations. Hardware details (i.e., number and type of GPUs) also seem missing from the Experimental Details section.

---

> ### Author Rebuttal · Authors · 2025-07-31
>
> Thank you very much for your thorough review and thoughtful suggestions. We have carefully addressed each concern, as detailed below.
>
> > Q1&W1: SHAPOOL’s generality and relation to attack-specific methods.
>
> (1) The section "Extension to Other Inference Attacks" refers to applying SHAPOOL to different MIA algorithms, rather than other types of inference attacks. Although our experiments primarily focus on MIAs, the proposed method can be extended to other inference attacks such as black-box property inference attacks (PIA) [1, 2], which aim to distinguish between statistical properties (e.g., $t_0$ vs. $t_1$). To adapt SHAPOOL for PIA, we can modify the training data for each pathway such that one SHAPOOL is trained on data satisfying $t_0$ and the other on $t_1$, replacing the shadow models for each property. We present the pseudo-code as follows. While we are currently developing the PIA pipeline and conducting experiments, the results may not be completed by the end of this rebuttal. Nonetheless, we will report these results in the final version of this paper.
>
> [1] Mahloujifar S, Ghosh E, Chase M. Property inference from poisoning[C]//2022 IEEE Symposium on Security and Privacy (SP). IEEE, 2022: 1120-1137.
> [2] Chaudhari H, Abascal J, Oprea A, et al. SNAP: Efficient extraction of private properties with poisoning[C]//2023 IEEE Symposium on Security and Privacy (SP). IEEE, 2023: 400-417.
>
>
> ********
> *Algorithm: SHAPOOL-based Property Inference Attacks*
> **Input**: Predefined values $t_0$ and $t_1$;  Attack dataset $D_{attack}$; Shadow dataset $\mathcal{D}$; Attack algorithm $\mathcal{A}$;
> Target model $M$; MoE model $f$;
> **Output**: Inferred property value $\hat{t} \in \{t_0, t_1\}$
> **For** $i \in \{0, 1\}$:
>  Initialize mapping matrix of $f_i$ via pathway-choice routing
>  **For** $j = 1$ **to** $m$:
>   Sample $D_{i,j}^1, D_{i,j}^2 \subset D$ such that $P(D_{i,j}^1) = P(D_{i,j}^2) = t_i$
>   Randomly select pathways $p_1$ and $p_2$ in $f_i$
>   Train $p_1$ and $p_2$ jointly using pathway regularization on $D_{i,j}^1$ and $D_{i,j}^2$
>  **End For**
>  Randomly select $n$ pre-trained pathways $S \subset f_i$
>  **For each** pathway $p \in S$:
>   Fine-tune $p$ via pathway alignment
>   Record confidence scores: $c_i \leftarrow p(D_{attack})$
>  **End For**
> **End For**
> Construct $\mathcal{A}$ using distributions:
>     $N_0 \sim c_0$, $N_1 \sim c_1$
> Collect confidence scores:
>     $\mathbf{c} \leftarrow M(D_{attack})$
> Compute probability:
>     $\ell_0 \leftarrow \mathcal{A}(\mathbf{c} \mid N_0)$,
>     $\ell_1 \leftarrow \mathcal{A}(\mathbf{c} \mid N_1)$
> Return: $\hat{t} \leftarrow \arg\max\limits_{i \in \{0,1\}} \ell_i $
> ********
>
> (2) By “coupled with specific algorithms,” we mean that prior works such as [10, 12, 13] are specific attack algorithm, not general-purpose shadow model construction techniques and thus cannot be easily reused across different attacks. In contrast, SHAPOOL focuses purely on shadow model construction, making it more flexible for integration into different attack pipelines.
>
> > Q2&W2: Description of the full ROC-curve comparison.
>
> SHAPOOL serves as a shadow model construction method that can be integrated into existing MIA frameworks such as LiRA and RMIA. Per your question, it yields ROC curves similar to the BASE versions under the same attack algorithm, with only some fluctuations and occasionally lower (less than 0.05) performance than BASE. Due to format constraints in this rebuttal, we are unable to enclose  the full ROC curves here but will provide them in the final version of this paper.
>
> > Q3: The performance of fewer SHAPOOLs.
>
> According to the ablation study (Figure 3b), we think that using fewer SHAPOOLs may lead to degraded performance and fall short of RMIA’s optimal results. Nonetheless, this is expected as the RMIA’s optimal results use unlimited number of individual shadow models, which incurs infeasible computational cost.
>
> > Q4&W2&W3: Performance under fixed compute budget.
>
> Take LiRA-online on CIFAR100 with ResNet18 as an example. A single SHAPOOL outperforms four individual shadow models **under the same compute budget**. Furthermore, four SHAPOOLs—at a cost equivalent to 16 individual models—nearly match the performance achieved by using 128 shadow models. These results suggest that SHAPOOL consistently outperforms LiRA across a range of fixed compute budgets. Supporting experimental results are provided below.
>
> Table 1: Performance (AUC/TF1) of LiRA-online with varying budgests on CIFAR100 (ResNet18)
> | #SHAPOOL (=SM)  | 1 (=4)   | 2 (=8)   | 3 (=12)  | 4 (=16)  |
> |---------------- |----------|----------|----------|----------|
> | BASE            | 0.90/0.34| 0.93/0.44| 0.94/0.46| 0.93/0.49|
> | SHAPOOL         | 0.92/0.42| 0.93/0.48| 0.93/0.50| 0.94/0.53|
>
>
> > W3: Clarification about experimental setup
>
> (1) Our goal is not to compare which MIA performs best under constrained settings, but to show how SHAPOOL improves the **efficiency** of shadow model construction. We use LiRA as a representative case and compare the performance of LiRA+BASE with that of LiRA+SHAPOOL.
> (2) While RMIA reports stable AUC with 4/8 models, we find that key metrics like TF1 can continue to improve with more models (e.g., TF1 increases from 0.44 to 0.53 but AUC stays 0.93–0.94 for 2~64 shadow models). To fairly compare with SHAPOOL’s upper bound, we use 64 models for RMIA.
>
> > W4: Reference and hardware details.
>
> Per your suggestion, we add citations for the Bhattacharyya coefficient [1] and the CINC10 dataset [2]. The hardware details are included in Appendix C.1, that is, NVIDIA RTX 3090 GPUs with 24GB of memory.
> [1] Bhattacharyya A. On a measure of divergence between two statistical populations defined by their probability distribution[J]. Bulletin of the Calcutta Mathematical Society, 1943, 35: 99-110.
> [2] Darlow L N, Crowley E J, Antoniou A, et al. Cinic-10 is not imagenet or cifar-10[J]. arXiv preprint arXiv:1810.03505, 2018.

---

> ### Comment · Reviewer_rfXF · 2025-08-04
>
> I thank the authors for responding to my comments!
>
> > Q1&W1: SHAPOOL’s generality and relation to attack-specific methods.
>
> (1) Thanks, I see! If the authors could make this clearer in the paper that would be great.
> (2) Ok I see. In that case, if I may, I think the point here is not that [10, 12, 13] are "coupled with specific algorithms" but rather that the focus of [10, 12, 13] is different from your work. Their focus is to come up with novel attack algorithms whereas yours solely focuses on shadow model construction, which are orthogonal to each other.
>
> > Q3: The performance of fewer SHAPOOLs.
>
> Sorry maybe my point was not clear. My point was that I would like to see a comparison between SHAPOOL and RMIA both using a limited compute budget (<= 4 shadow models) kind of like the only you have already sent for SHAPOOL vs BASE for LiRA. Would this be possible? You don't have to send over results now but if you could commit to it/explain why this cannot be done that would be great. Thanks!
>
> > Q4&W2&W3: Performance under fixed compute budget.
>
> Wow these results look really promising thanks! Just a minor question when comparing with LiRA here do you fix the variance for all samples? I think the LiRA paper originally reported that in low reference model settings, fixed variance outperforms per-sample variance substantially (see Figure 17). You don't have to send over the new results (if any) now but if you could make sure that this is clear in the final version of the paper that would be great.
>
> > W3: Clarification about experimental setup
>
> I understand where the authors are coming from but RMIA's main contribution was specifically in the low reference model setting. Therefore I still believe that it is unfair that the authors only compare with RMIA when using many reference models. In any case isn't it an advantageous for the authors if for e.g., for 2 reference models RMIA's AUC ~ SHAPOOL's AUC but SHAPOOL's TF1 substantially outperforms RMIA? We can discuss this point in greater detail but I strongly believe that it is important to compare against the appropriate MIA for the setting. My suggestion is for the authors to report RMIA for Table 1 since RMIA is the best for "constrained resource" setting. Then Table 3 can remain as-is since Table 1 would already have evaluated RMIA under resource-constrained setting. Happy to hear the authors' thoughts on this.
>
> I thank the authors for sincerely addressing the comments I have raised and will improve my score to 4. The only remaining concern I have pertains to the comparison with RMIA above and if the authors can commit to addressing this I will be inclined to improve my score further. Thanks!

---

> > ### Author Response · Authors · 2025-08-04
> >
> > Thank you very much for your valuable comments, particularly regarding the constrained resource scenario.
> > We apologize for the confusion regarding Q3. Our original work pays more attention to the unconstrained resource setting and thus adopts LiRA as a representative method in this context. To address your concern on Q3 and W3 under the constrained resource setting (especially a direct comparison with RMIA), we just finished the experiment and reported results under RMIA (online version) using only two reference models and one shadow pool. The results (AUC/TF1/TF01) are as follows: RMIA+BASE: 0.93/0.44/0.21 and RMIA+SHAPOOL: 0.93/0.43/0.26. From these results, by using RMIA under constrained resources, our SHAPOOL achieves (almost) the same AUC and TF1 while outperforms BASE in terms of TF01.

---

> > > ### Comment · Reviewer_rfXF · 2025-08-05
> > >
> > > Thank you for getting back to me with the updated results for RMIA! Yes it is as what I had expected. Assuming the results from these discussions are included in the final submission, I will improve my score to 5. Good luck!

---

> > > > ### Author Response · Authors · 2025-08-06
> > > >
> > > > Thank you very much for taking the time to review our submission and for providing thoughtful feedback. We are especially grateful that our clarifications met your expectations and contributed to the score improvement. Your constructive comments have been truly valuable in helping us strengthen this work. Thank you once more for your great insights!

---

### Official Review · Reviewer_Y9em · 2025-06-30

**Clarity:** 3
**Significance:** 2
**Originality:** 3
**Rating:** 4
**Confidence:** 5

**Summary:**

This paper proposes SHAPOOL, a shared shadow model training framework based on MoE, to reduce the computational cost of inference attacks through joint training with subnetwork sharing. The paper designs three main modules Pathway-choice Routing,  Pathway Regularization, and  Pathway Alignment to address the real-world challenges of MoE, and experiments show that the framework is able to have a low computational overhead against membership inference attacks.

**Questions:**

- Why did the performance comparison in the experiment only compare BASELINE and not the other  related methods?
- How many times has the experiment been run and how stable is it?
- Considering that the construction of shadow models is actually very dependent on training data, is the performance of the model maintained when only a small amount of data is available?

**Ethical Concerns:**

["NO or VERY MINOR ethics concerns only"]

**Final Justification:**

The authors have taken most of my concerns into consideration. However, the W1 and W2 still need improvement.

**Limitations:**

Yes

**Quality:**

3

**Strengths And Weaknesses:**

Strengths：

- Although the use of MoE to enhance the performance of shadowing models is rather intuitive, it is also a way to bring new research perspectives to existing studies.
- The article has a logical narrative, relatively adequate experiments, and open source code.

Weaknesses：

- Many of the statements in the main text reflect that the paper focuses on inference attacks rather than specifically on MIA, but only MIA is used in terms of both framework design and experimental evaluation. Therefore, the paper lacks focus in some of the narratives and lacks experimental validation of other inference attacks such as FIA (in fact, other inference attack methods may not require the use of shadow models, which also leads to a conflict with the concept of inference attack referred to in the article).
- In addition to the intuitive introduction of the MoE approach, the design of the three core modules is actually rather intuitive, so I am pessimistic about the real contribution of these approaches.
- An obvious flaw in the experimental design is that it only compares BASELINE and lacks comparisons with other shadow model building methods. As far as I know this part of the work is very extensive, so did the authors forget to compare with these or was the performance limited?
- The experiment results are missing statistical data (e.g., variance, etc.).
- The data font sizes in Figures 3 and 6 are too small and can be adjusted subsequently.

---

> ### Author Rebuttal · Authors · 2025-07-31
>
> We sincerely appreciate your time of reviewing our paper and insightful suggestions. We now address your concerns in details below.
>
> > Q1&W3: About baseline and related methods.
>
> We consider the BASE method as an ideal baseline for training shadow models, as it independently trains shadow models with the same architecture and same data distribution as the target. As such, BASE is a natural and widely adopted approach, especially in black-box MIAs. The only disadvantage is its high computational cost, which is the main motivation of this paper to train similar quality of shadow models with much lower cost.
>
> Regarding alternative shadow model construction strategies, naive methods such as simple model augmentation have shown limited effectiveness in our context. The work [R1] proposed a distilled construction strategy, where shadow models are generated from a pre-trained model via distillation. However, as shown in the table below, this approach yields suboptimal performance—whether using 64 or 128 distilled models.
>
> Table 1: RMIA-online version on CIFAR100 and ResNet18.
> |Method|AUC|TPR@FPR=1%|Cost(h)|
> |------|---|----------|-------|
> |BASE|0.94|0.53|16.0|
> |Distilled(64)|0.85|0.17|1.85|
> |Distilled(128)|0.83|0.21|3.45|
> |SHAPOOL|0.93|0.53|4.0|
>
> [R1] Ye J, Maddi A, Murakonda S K, et al. Enhanced membership inference attacks against machine learning models[C]//Proceedings of the 2022 ACM SIGSAC conference on computer and communications security. 2022: 3093-3106.
>
>
> > Q2&W4: Experiments setup and stability.
>
> We ran each experiment five times with different random seeds, as described in Section 5.1. To address your concern, we update Tables 1 to include standard deviation as below. Compared to BASE, our results show the same or slightly higher fluctuations in AUC and FPR.
>
> Table 1 - LiRA-offline attack
> |Dataset |Method  |ResNet18 AUC|TF1 |Cost|VGG16 AUC|TF1 |Cost|WRN28-10 AUC|TF1 |Cost|
> |--------|--------|------------|-----|----|----------|-----|----|-------------|-----|----|
> |CIFAR100|BASE    |0.73±0.00   |0.22±0.03|0.5±0.10|0.69±0.01|0.16±0.01|0.4±0.06|0.76±0.02|0.30±0.01|2.4±0.11|
> |        |SHAPOOL |0.82±0.01   |0.32±0.04|0.4±0.10|0.76±0.01|0.17±0.05|0.5±0.08|0.81±0.02|0.27±0.04|1.9±0.06|
> |        |Δ       |+0.09       |+0.10     |-        |+0.07     |=       |-       |+0.05     |-0.03    |-       |
> |CIFAR10 |BASE    |0.51±0.01   |0.06±0.00|0.5±0.13|0.56±0.01|0.04±0.00|0.4±0.08|0.55±0.02|0.08±0.01|2.2±0.06|
> |        |SHAPOOL |0.63±0.02   |0.09±0.02|0.5±0.09|0.64±0.02|0.09±0.02|0.4±0.14|0.60±0.02|0.08±0.02|1.6±0.14|
> |        |Δ       |+0.12       |+0.03     |-        |+0.08     |+0.05   |-       |+0.05     |=        |-       |
>
> Table 1 - LiRA-online attack
> |Dataset |Method  |ResNet18 AUC|TF1 |Cost|VGG16 AUC|TF1 |Cost|WRN28-10 AUC|TF1 |Cost|
> |--------|--------|------------|-----|----|----------|-----|----|-------------|-----|----|
> |CIFAR100|BASE    |0.90±0.00   |0.34±0.01|1.0±0.10|0.86±0.00|0.25±0.03|0.9±0.04|0.90±0.01|0.36±0.03|3.6±0.11|
> |        |SHAPOOL |0.92±0.01   |0.42±0.04|1.0±0.13|0.88±0.01|0.25±0.02|0.9±0.15|0.91±0.01|0.42±0.04|3.4±0.08|
> |        |Δ       |+0.02       |+0.08     |-        |+0.02     |=       |-       |=         |+0.06    |-       |
> |CIFAR10 |BASE    |0.69±0.00   |0.11±0.01|1.0±0.11|0.67±0.00|0.07±0.01|0.9±0.08|0.67±0.02|0.09±0.01|4.4±0.07|
> |        |SHAPOOL |0.71±0.03   |0.11±0.03|1.0±0.13|0.69±0.02|0.10±0.02|1.2±0.13|0.69±0.02|0.12±0.02|4.5±0.06|
> |        |Δ       |+0.02       |=         |-        |+0.02     |+0.03   |-       |+0.02     |+0.03    |-       |
>
>
>
> > Q3: Impact of the amount of available shadow dataset.
>
> We agree with you that the amount of available training data can affect performance, as discussed in Appendix D. Since our method is based on an MoE model with more parameters than an individual model, sufficient training data can achieve good performance. Nonetheless, only when the available data is extremely limited, for instance, less than 1.2× the amount used for an individual shadow model, the performance of SHAPOOL may degrade, as shown in Table 6 of Appendix C.3.
>
>
> > W1: SHAPOOL’s generality
>
> Although our experiments primarily focus on MIAs, SHAPOOL can be extended to other attacks such as black-box property inference attacks (PIA) [1, 2], which aim to distinguish statistical properties (e.g., between $t_0$ and $t_1$). To adapt SHAPOOL for PIA, we can modify the training data of each pathway so that one SHAPOOL is trained on data satisfying $t_0$ and the other on $t_1$, replacing the shadow models for each property, as shown in the pseudo-code below. While we are currently developing the PIA pipeline and conducting experiments, the results may not be completed by the end of this rebuttal. Nonetheless, we will report these results in the final version of this paper.
>
> [1] Mahloujifar S, Ghosh E, Chase M. Property inference from poisoning[C]//2022 IEEE Symposium on Security and Privacy (SP). IEEE, 2022: 1120-1137.
> [2] Chaudhari H, Abascal J, Oprea A, et al. SNAP: Efficient extraction of private properties with poisoning[C]//2023 IEEE Symposium on Security and Privacy (SP). IEEE, 2023: 400-417.
>
>
> ********
> *Algorithm: SHAPOOL-based Property Inference Attacks*
> **Input**: Predefined values $t_0$ and $t_1$;  Attack dataset $D_{attack}$; Shadow dataset $\mathcal{D}$; Attack algorithm $\mathcal{A}$;
> Target model $M$; MoE model $f$;
> **Output**: Inferred property value $\hat{t} \in \{t_0, t_1\}$
> **For** $i \in \{0, 1\}$:
>  Initialize mapping matrix of $f_i$ via pathway-choice routing
>  **For** $j = 1$ **to** $m$:
>   Sample $D_{i,j}^1, D_{i,j}^2 \subset D$ such that $P(D_{i,j}^1) = P(D_{i,j}^2) = t_i$
>   Randomly select pathways $p_1$ and $p_2$ in $f_i$
>   Train $p_1$ and $p_2$ jointly using pathway regularization on $D_{i,j}^1$ and $D_{i,j}^2$
>  **End For**
>  Randomly select $n$ pre-trained pathways $S \subset f_i$
>  **For each** pathway $p \in S$:
>   Fine-tune $p$ via pathway alignment
>   Record confidence scores: $c_i \leftarrow p(D_{attack})$
>  **End For**
> **End For**
> Construct $\mathcal{A}$ using distributions:
>     $N_0 \sim c_0$, $N_1 \sim c_1$
> Collect confidence scores:
>     $\mathbf{c} \leftarrow M(D_{attack})$
> Compute probability:
>     $\ell_0 \leftarrow \mathcal{A}(\mathbf{c} \mid N_0)$,
>     $\ell_1 \leftarrow \mathcal{A}(\mathbf{c} \mid N_1)$
> Return: $\hat{t} \leftarrow \arg\max\limits_{i \in \{0,1\}} \ell_i $
> ********
>
> > W2: Design of three modules.
>
> We agree that the modules are based on intuitive principles; however, their novelty lies in adapting MoE for shared shadow model construction through minimal yet effective modifications. As demonstrated in our ablation study (Figure 5 and Table 4), each module contributes meaningfully to the final attack performance. Moreover, the simplicity of our design facilitates easy integration, while achieving comparable or even superior performance (Tables 1 and 2) compared to traditional shadow model construction methods.
>
>
> > W5: Experimental figure settings.
>
> We will revise Figures 3 and 6 to improve readability in the final version.

---

> > ### Comment · Reviewer_Y9em · 2025-08-06
> >
> > Thank you for your response. I will improve my score. Good luck!

---

> > > ### Author Response · Authors · 2025-08-06
> > >
> > > We sincerely appreciate your time and effort in reviewing our submission. We are especially pleased that our rebuttal successfully addressed your concerns and got an improved score. Thank you once again for your thoughtful review and encouraging comments!

---

> ### Author Response · Authors · 2025-08-06
>
> Dear Reviewer,
>
> Thank you again for your valuable time and detailed review. We have posted a rebuttal where we sought to address your concerns with new empirical evidence and further clarifications.
>
> We would be grateful to know if our response has helped clarify these points, and happy to discuss any remaining questions you may have.
>
> Thank you once more for your insights.
>
> Best Regards.

---

### Official Review · Reviewer_ru5X · 2025-07-02

**Clarity:** 4
**Significance:** 4
**Originality:** 4
**Rating:** 5
**Confidence:** 3

**Summary:**

This paper aims to tackle the problem of efficient membership inference attacks—specifically, how to reduce the number of shadow models required for such attacks. To this end, the paper proposes a novel shadow pool training framework, SHAPOOL, which constructs multiple shared models and trains them jointly within a single process. In addition, the authors introduce three novel modules—path-choice routing, pathway regularization, and pathway alignment—to ensure that the shared models closely resemble independently trained models and can serve as effective substitutes.

**Questions:**

See Weakness above.

**Ethical Concerns:**

["NO or VERY MINOR ethics concerns only"]

**Final Justification:**

My concerns have been well addressed. Therefore, I keep my score.

**Limitations:**

Yes.

**Paper Formatting Concerns:**

No.

**Quality:**

4

**Strengths And Weaknesses:**

Strengths:
- The studied problem is interesting and important to the community.
- The motivations for the studied problem and the proposed solution are both very sufficient.
- The writing and the structure of the paper are good.
- My expertise is not in the field. However, the paper provides necessary background information for me to understand the problem setting.
- The proposed method makes sense in general.

Weaknesses:
- Experiments: The existing experiments are conducted Cifar10, Cifar100, and CINIC10. Could you also provide some experiments on datasets of large image size, e.g., ImageNet-Subset?

---

> ### Author Rebuttal · Authors · 2025-07-31
>
> Thank you very much for your time and valuable comments, most of which are positive and encouraging. Due to very tight rebuttal schedule, for your suggestion to conduct experiments on larger-image datasets, we use the TinyImageNet [1] derived from ImageNet and ResNet18 under the unconstrained setting for RMIA. The results, presented below, provide further support for our approach. We will consider using even larger-image datasets in the final version of this paper.
>
> [1] Ya Le and X Yang. Tiny ImageNet Visual Recognition Challenge. 2015.
>
> Table: Performance and cost of RMIA on TinyImageNet.
>
> |       |**RMIA-OFFLINE**||||**RMIA-ONLINE**||||
> |-------|---|---|----|----|---|---|----|----|
> |       |AUC|TF1|TF01|COST|AUC|TF1|TF01|COST|
> |BASE   |0.99|0.95|0.86|28.1h|0.99|0.94|0.84|61.9h|
> |SHAPOOL|0.99|0.94|0.87|7.9h |0.99|0.95|0.88|16.1h|
> |Δ      |=   |=   |=   |↓72% |=   |=   |+0.04|↓74% |

---

### Decision · Program_Chairs · 2025-09-17

**Decision:**

Accept (poster)

**Comment:**

This paper tackles an important computational bottleneck in common strategies for inference attacks--the use of independently trained shadow models. The reviewers agree that the use of a mixture-of-experts architecture is nice and novel. The authors have also addressed many questions initially raised by the reviewers.